# Cross-Scale Internal Graph Neural Network for Image Super-Resolution

**Shangchen Zhou**[1]     **Jiawei Zhang**[2]     **Wangmeng Zuo**[3]     **Chen Change Loy**[1]*
[1]Nanyang Technological University     [2]SenseTime Research     [3]Harbin Institute of Technology
`{s200094,ccloy}@ntu.edu.sg`   `zhangjiawei@sensetime.com`   `wmzuo@hit.edu.cn`
`https://github.com/sczhou/IGNN`

## Abstract

Non-local self-similarity in natural images has been well studied as an effective prior in image restoration. However, for single image super-resolution (SISR), most existing deep non-local methods (e.g., non-local neural networks) only exploit similar patches within the same scale of the low-resolution (LR) input image. Consequently, the restoration is limited to using the same-scale information while neglecting potential high-resolution (HR) cues from other scales. In this paper, we explore the cross-scale patch recurrence property of a natural image, i.e., similar patches tend to recur many times across different scales. This is achieved using a novel cross-scale internal graph neural network (IGNN). Specifically, we dynamically construct a cross-scale graph by searching $k$-nearest neighboring patches in the downsampled LR image for each query patch in the LR image. We then obtain the corresponding $k$ HR neighboring patches in the LR image and aggregate them adaptively in accordance to the edge label of the constructed graph. In this way, the HR information can be passed from $k$ HR neighboring patches to the LR query patch to help it recover more detailed textures. Besides, these internal image-specific LR/HR exemplars are also significant complements to the external information learned from the training dataset. Extensive experiments demonstrate the effectiveness of IGNN against the state-of-the-art SISR methods including existing non-local networks on standard benchmarks.

## 1   Introduction

The goal of single image super-resolution (SISR) [9] is to recover the sharp high-resolution (HR) counterpart from its low-resolution (LR) observation. Image SR is an ill-posed problem, since there are multiple HR solutions for a LR input. To solve this inverse problem, many convolutional neural networks (CNNs) [6, 38, 17, 23, 43, 13, 5] have been proposed to capture useful priors by learning mappings between LR and HR images. While immense performance has been achieved, learning from external training data solely still falls short in recovering detailed textures for specific images, especially when the up-scaling factor is large.

Apart from exploiting external paired data, internal image-specific information [30] has also been widely studied in image restoration. Some classical non-local methods [2, 4, 25, 10] have shown the values of capturing correlation among non-local self-similar patches for improving the restoration quality. However, convolutional operations are not able to capture such patterns due to the locality of convolutional kernels. Though the receptive fields are large in the deep networks, some long-range dependencies still cannot be well maintained. Inspired by the classical non-local means method [2], non-local neural networks [37] are proposed to capture long-range dependencies for video classification. Non-local neural networks are thereafter introduced to image restoration tasks [24, 44].

---

These methods, in general, perform self-attention weighting of full connection among positions in the features. Besides non-local neural networks, the neural nearest neighbors network [29] and graph-convolutional denoiser network [36] have been proposed to aggregate $k$ nearest neighboring patches for image restoration. However, all these methods only exploit correlations of recurrent patches within the same scale, without harvesting any high-resolution information. Different from image denoising, the aggregation of multiple similar patches at the same scale (subpixel misalignments) only improves the performance slightly for SR.

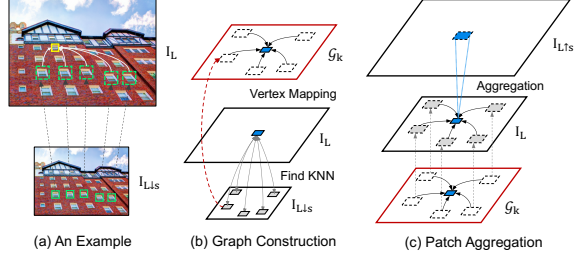

**Figure 1:** Example of patch recurrence across scales of a single image (a), and illustration of Graph Construction (b) and Patch Aggregation (c) in the image domain. The $I_L$ and $I_{L\downarrow s}$ are input LR image and its $s$ downsampled counterpart. $\mathcal{G}_k$ is the constructed cross-scale graph and $I_{L\uparrow s}$ is the patch aggregated result with LR$_{\uparrow}s$ scale.

The proposed the cross-scale internal graph neural network (IGNN) is inspired by the traditional self-example based SR methods [7, 3, 14]. Our IGNN is based on cross-scale patch recurrence property verified statistically in [46, 9, 28] that patches in a natural image tend to recur many times across scales. An illustrative example is shown in Figure 1 (a). Given a query patch (yellow square) in the LR image $I_L$, many similar patches (solid-marked green squares) can be found in the downsampled image $I_{L\downarrow s}$. Thus the corresponding HR patches (dashed-marked green squares) in the LR image $I_L$ can also be obtained. Such cross-scale patches provide an indication of what the unknown HR patches of the query patch might look like. The cross-scale patch recurrence property is previously utilized as example-based SR constraints to estimate a HR image [9, 40] or a SR kernel [28].

In this paper, we model this internal correlations between cross-scale similar patches as a graph, where every patch is a vertex and the edge is similarity-weighted connection of two vertexes from two different scales. Based on this graph structure, we then present our IGNN to process this irregular graph data and explore cross-scale recurrence property effectively. Instead of using this property as constraints [9, 28], IGNN intrinsically aggregates HR patches using the proposed graph module, which includes two operations: graph construction and patch aggregation. More specifically, as shown in Figure 1 (b)(c), we first dynamically construct a cross-scale graph $\mathcal{G}_k$ by searching $k$-nearest neighboring patches in the downsampled LR image $I_{L\downarrow s}$ for each query patch in the LR image $I_L$. After mapping the regions of $k$ neighbors from $I_{L\downarrow s}$ to $I_L$ scale, the constructed cross-scale graph $\mathcal{G}_k$ can provide $k$ LR/HR patch pairs for each query patch. In $\mathcal{G}_k$, the vertexes are the patches in LR image $I_L$ and their $k$ HR neighboring patches and the edges are correlations of these matched LR/HR patches. Inspired by Edge-Conditioned Convolution [31], we formulate an edge-conditioned patch aggregation operation based on the graph $\mathcal{G}_k$. The operation aggregates $k$ HR patches conditioned on edge labels (similarity of two matched patches). Different from previous non-local methods that explore and aggregate neighboring patches at the same scale, we search for similar patches at the downsampled LR scale but aggregate HR patches. It allows our network to perform more efficiently and effectively for SISR.

The proposed IGNN obtains $k$ image-specific LR/HR patch correspondences as helpful complements to the external information learned from a training dataset. Instead of learning a LR-to-HR mapping only from external data as other SR networks do, the proposed IGNN makes full use of $k$ most likely HR counterparts found from the LR image itself to recover more detailed textures. In this way, the ill-posed issue in SR can be alleviated in IGNN. We thoroughly analyze and discuss the proposed graph module via extensive ablation studies. The proposed IGNN performs favorably against state-of-the-art CNN-based SR baselines and existing non-local neural networks, demonstrating the usefulness of cross-scale graph convolution for image super-resolution.

## 2   Methodology

In this section, we start by briefly reviewing the general formulation of some previous non-local methods. We then introduce the proposed cross-scale graph aggregation module (GraphAgg) based on graph message aggregation methods [8, 11, 19, 42, 31]. Built on GraphAgg module, we finally present our cross-scale internal graph neural network (IGNN).

## 2.1 Background of Non-local Methods for Image Restoration

Non-local aggregation strategy has been widely applied in image restoration. Under the assumption that similar patches frequently recur in a natural image, many classical methods, e.g., non-local means [2] and BM3D [4], have been proposed to aggregate similar patches for image denoising. With the development of deep neural network, the non-local neural networks [37, 24, 44] and some $k$-nearest neighbor based networks [21, 29, 36] are proposed for image restoration to explore this non-local self-similarity strategy. For these non-local methods that consider similar patch aggregation, the aggregation process can be generally formulated as:

$$\boldsymbol{Y}^i = \frac{1}{\delta_i(\boldsymbol{X})} \sum\nolimits_{j \in \mathcal{S}_i} \boldsymbol{C}(\boldsymbol{X}^i, \boldsymbol{X}^j) \boldsymbol{Q}(\boldsymbol{X}^j), \;\; \forall i, \tag{1}$$

where $\boldsymbol{X}^i$ and $\boldsymbol{Y}^i$ are the input and output feature patch (or element) at $i$-th location (aggregation center), and $\boldsymbol{X}^i$ is also the query item in Eq. (1). $\boldsymbol{X}^j$ is the $j$-th neighbors included in the neighboring feature patch set $\mathcal{S}_i$ for $i$-th location. The $\boldsymbol{Q}(\cdot)$ transforms the input $\boldsymbol{X}$ to the other feature space. As for $\boldsymbol{C}(\cdot, \cdot)$, it computes an aggregation weights for transformed neighbors $\boldsymbol{Q}(\boldsymbol{X}^j)$ and the more similar patch relative to $\boldsymbol{X}^i$ should have the larger weight. The output is finally normalized by a factor $\delta_i(\boldsymbol{X})$, i.e., $\delta_i(\boldsymbol{X}) = \sum_{j \in \mathcal{S}_i} \boldsymbol{C}(\boldsymbol{X}^i, \boldsymbol{X}^j)$.

The above aggregation can be treated as a GNN if we treat the feature patches and weighted connections as vertices and edges respectively. The non-local neural networks [37, 24, 44] actually model a fully-connected self-similarity graph. They estimate the aggregation weights between the query item $\boldsymbol{X}_i$ and all the spatially nearby patches $\boldsymbol{X}_j$ in a $d \times d$ window (or within the whole features). To reduce the memory and computational costs introduced by the above dense connection, some $k$-nearest neighbor based networks, e.g., GCDN [36] and N$^3$Net [29], only consider $k$ ($k \ll d^2$) most similar feature patches for aggregation and treat them as the neighbors in $\mathcal{S}_i$ for every query $\boldsymbol{X}^i$. For all the above mentioned non-local methods, the aggregated neighboring patches are all in the same scale of the query and no HR information is incorporated, thus leading to a limited performance improvement for SISR. In [9, 46, 28], Irani *et al.* notice that patch recurrence also exists across the different scales. They explore these cross-scale recurrent LR/HR pairs as example-based constraints to recover the HR images [9, 40] or to estimate the SR kernels [28] from the LR images.

## 2.2 Cross-Scale Graph Aggregation Module

For the aforementioned methods [2, 4, 24, 36, 29], the patch size of the aggregated feature patches is the same as the query one. Even though it works well for image denoising, it fails to incorporate high-resolution information and only provides limited improvement for SR. Based on the patch recurrency property [46] that similar patches will recur in different scales of nature image, we propose a cross-scale internal graph neural network (IGNN) for SISR. An example of patch aggregation in image domain is shown in Figure 1. For each query patch (yellow square) in $I_L$, we search for the $k$ most similar patches (solid-marked squares) in the downsampled image $I_{L\downarrow s}$. we then aggregate their $k$ HR corresponding patches (dashed-marked squares) in $I_L$.

The connections between cross-scale patches can be well constructed as a graph, where every patch is a vertex and edge is a similarity-weighted connection of two vertices from two different scales. To exploit the information of HR patches for SR, we propose a cross-scale graph aggregation module (GraphAgg) to aggregate HR patches in feature domain. As shown in Figure 2, the GraphAgg includes two operations: Graph Construction and Patch Aggregation.

**Graph Construction:** We first downsample the input LR image $I_L$ by a factor of $s$ using the widely used Bicubic operation. The downsampled image is denoted as $I_{L\downarrow s}$, where the downsampling ratio $s$ is equal to the desired SR up-scaling factor. Thus the found $k$ neighboring feature patches in graph $\mathcal{G}_k$ are the same size as the desired HR feature patch. However, we find that the downsampling ratio $s = 2$ is better than $s = 4$ for up-scaling $\times 4$ since it is much more difficult to find accurate $\times 4$ neighbors directly. Thus, we search for $\times 2$ neighbors instead and obtain $\times 2$ HR features $F_{L\uparrow s}$ aggregated by GraphAgg. We then concatenate $F_{L\uparrow s}$ with features after the first $\times 2$ upsample *PixelShuffle* operation[1].

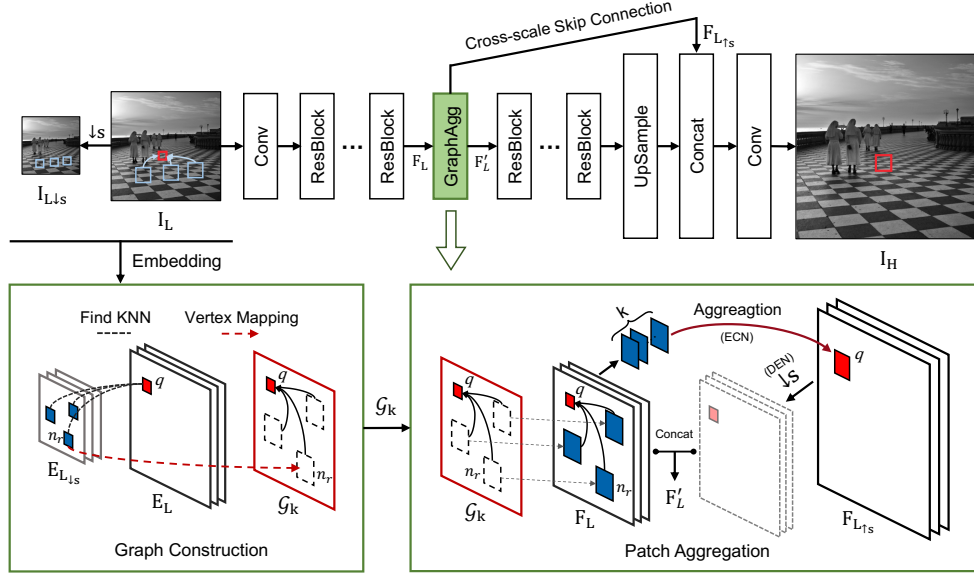

**Figure 2:** An illustration of the proposed the cross-scale Internal Graph Neural Network (IGNN) and the Cross-Scale Graph Aggregation module (GraphAgg). The GraphAgg includes two operations: Graph Construction and Patch Aggregation. A cross-scale graph $\mathcal{G}_k$ is constructed by Graph Construction. Taking $\mathcal{G}_k$ as input, the HR features $F_{L\uparrow s}$ and the enriched LR features $F'_L$ are obtained by Patch Aggregation, which enables our network to take advantage of internal HR information to recover more details. The skip connection across different scales passes the HR features $F_{L\uparrow s}$ to enrich subsequent upsampled features.

To obtain the $k$ neighboring feature patches, we first extract embedded features $E_L$ and $E_{L\downarrow s}$ by the first three layers of VGG19 [32] from $I_L$ and $I_{L\downarrow s}$, respectively. Following the notion of block matching in classical non-local methods [2, 4, 25, 10], for a $l \times l$ query feature patch $E_L^{q,l}$ in $E_L$, we find k $l \times l$ nearest neighboring patches $E_{L\downarrow s}^{n_r,l}, r = \{1, ..., k\}$ in $E_{L\downarrow s}$ according to the Euclidean distance between the query feature patch and neighboring ones. Then, we can get the $ls \times ls$ HR feature patch $E_L^{n_r,ls}$ corresponding to $E_{L\downarrow s}^{n_r,l}$ in $E_L$. We mark this process with a dashed red line in Figure 2, denoted as Vertex Mapping.

Consequently, a cross-scale $k$-nearest neighbor graph $\mathcal{G}_k(\mathcal{V}, \mathcal{E})$ is constructed. $\mathcal{V}$ is the patch set (vertices in graph) including a LR patch set $\mathcal{V}^l$ and a HR neighboring patch set $\mathcal{V}^{ls}$, where the size of $\mathcal{V}^l$ equals to number of LR patches in $E_L$. Set $\mathcal{E}$ is the correlation set (edges in graph) with size $|\mathcal{E}| = |\mathcal{V}^l| \times k$, which indicates $k$ correlations in $\mathcal{V}^{ls}$ for each LR patch in $\mathcal{V}^l$. The two vertices of each edge in this cross-scale graph $\mathcal{G}_k$ are LR and HR feature patches, respectively. To measure the similarity of query $q$ and the $r$-th neighbor $n_r$, we define the edge label as the difference between the query feature patch $E_L^{q,l}$ and neighboring patch $E_{L\downarrow s}^{n_r,l}$, i.e., $\mathcal{D}^{n_r \to q} = E_L^{q,l} - E_{L\downarrow s}^{n_r,l}$. It will be used to estimate aggregation weights in the following Patch Aggregation operation.

We search similar patches from $E_{L\downarrow s}$ rather than $E_L$, hence our searching space is $s^2$ times smaller than previous non-local methods. Unlike the fully-connected feature graph in non-local neural networks [24], we only select $k$ nearset HR neighbors for aggregation, which further leads to a more efficient network. Following the previous non-local methods [2, 4, 24], we also design a $d \times d$ searching window in $E_{L\downarrow s}$, which is centered with the position of the query patch in the downsampled scale. As verified statistically in [46, 9, 28], there are abundant cross-scale recurring patches in the whole single image. Our experiments show that searching for $k$ HR parches from a window region is sufficient for the network to achieve the desired performance.

**Patch Aggregation:** Inspired by Edge-Conditioned Convolution (ECC) [31], we aggregate $k$ HR neighbors in graph $\mathcal{G}_k$ weighted on the edge labels $\mathcal{D}^{n_r \to q}$. Our Patch Aggregation reformulates the general non-local aggregation Eq. (1) as:

$$F_{L\uparrow s}^{q,ls} = \frac{1}{\delta_q(F_L)} \sum_{n_r \in \mathcal{S}_q} \exp\left(\mathcal{F}_\theta\left(\mathcal{D}^{n_r \to q}\right)\right) F_L^{n_r,ls}, \quad \forall q, \tag{2}$$

where $F_L^{n_r,ls}$ is the $r$-th neighboring $ls \times ls$ HR feature patch from GraphAgg module input $F_L$ and $F_{L\uparrow s}^{q,ls}$ is the output HR feature patch at the query location. And the *patch2img* [29] operator is used to transform the output feature patches into the output feature $F_{L\uparrow s}$. We propose to use an adaptive Edge-Conditioned sub-network (ECN), i.e., $\mathcal{F}_\theta \left( \mathcal{D}^{n_r \rightarrow q} \right)$, to estimate the aggregation weight for each neighbor according to $\mathcal{D}^{n_r \rightarrow q}$, which is the feature difference between the query patch and neighboring patch from the embedded feature $E_L$. We use $\exp(\cdot)$ to denote the exponential function and $\delta_q(F_L) = \sum_{n_r \in \mathcal{S}_q} \exp\left(\mathcal{F}_\theta \left( \mathcal{D}^{n_r \rightarrow q} \right)\right)$ to represent the normalization factor. Therefore, Eq. (1) defines an adaptive edge-conditioned aggregation utilizing the sub-network ECN. By exploiting edge labels (i.e., $\mathcal{D}^{n_r \rightarrow q}$), GraphAgg aggregates $k$ HR feature patches in a robust and flexible manner.

To further utilize $F_{L\uparrow s}$, we use a small Downsampled-Embedding sub-network (DEN) to embed it to a feature with the same resolution as $F_L$ and then concatenate it with $F_L$ to get $F_L'$. Then $F_L'$ is used in subsequent layers of the network. Note that the two sub-networks ECN and DEN in Patch Aggregation are both very small networks containing only three convolutional layers, respectively. Please see Figure 2 for more details.

**Adaptive Patch Normalization:** We observe that the obtained $k$ HR neighboring patches have some low-frequency discrepancy, e.g., color, brightness, with the query patch. Besides the adaptive weighting by edge-conditioned aggregation, we propose an Adaptive Patch Normalization (AdaPN), which is inspired by Adaptive Instance Normalization (AdaIN) [15] for image style transfer, to align the $k$ neighboring patches to the query one. Let us denote $F_L^{q,l,c}$ and $F_L^{n_r,ls,c}$ as the $c$-th channel of features of the query patch and $r$-th HR neighboring patch in $F_L$, respectively. The $r$-th normalized neighboring patch $F_L^{n_r,ls,c}$ by AdaPN is formulated as:

$$\text{AdaPN}(F_L^{n_r,ls,c}|F_L^{q,l,c}) = \sigma(F_L^{q,l,c}) \left( \frac{F_L^{n_r,ls,c} - \mu(F_L^{n_r,ls,c})}{\sigma(F_L^{n_r,ls,c})} \right) + \mu(F_L^{q,l,c}), \qquad (3)$$

where $\sigma$ and $\mu$ are the mean and standard deviation. By aligning the mean and variance of the each neighboring patch features with those of the query patch one, AdaPN transfers the low-frequency information of the query to the neighbors and keep their high-frequency texture information unchanged. By eliminating the discrepancy between query patch and $k$ neighbor patches, the proposed AdaPN benefits the subsequent feature aggregation.

## 2.3  Cross-Scale Internal Graph Neural Network

As shown in Figure 2, we build the IGNN based on GraphAgg. After the GraphAgg module, a final HR feature $F_{L\uparrow s}$ is obtained. With a skip connection across different scales, the rich HR information in aggregated HR feature $F_{L\uparrow s}$ is passed directly from the middle to the late position in the network. This mechanism allows the HR information to help the network in generating outputs with more details. Besides, the enriched intermediate feature $F_L'$ is obtained by concatenating the input feature $F_L$ and the downsampled-embedded feature from $F_{L\uparrow s}$ using sub-network DEN. It is then fed into the subsequent layers of the network, enabling the network to explore more cross-scale internal information.

Compared to the previous non-local networks [21, 29, 24, 44] for image restoration that only exploit self-similarity patches with the same LR scale, the proposed IGNN exploits internal recurring patches across different scales. Benefits from the GraphAgg module, IGNN obtains $k$ internal image-specific LR/HR feature patches as effective HR complements to the external information learned from a training dataset. Instead of learning a LR-to-HR mapping only from external data as other CNN SR networks do, IGNN takes advantage of $k$ most likely HR counterparts to recover more detailed textures. By $k$ LR/HR exemplars mining, the ill-posed issue of SR can be mitigated in the IGNN.

To show the effectiveness of our GraphAgg module, we choose the widely used EDSR [23] as our backbone network, which contains 32 residual blocks. The proposed GraphAgg module is only used once in IGNN and it is inserted after the 16th residual block.

In Graph Construction, we use the first three layers of the VGG19 [32] with fixed pre-trained parameters to embed image $I_L$ and $I_{L\downarrow s}$ to $E_L$ and $E_{L\downarrow s}$, respectively. In Graph Aggregation, both adaptive edge-conditioned network and downsample-embedding network are small network with three convolutional layers. More detailed structures are provided in the supplementary material.

# 3 Experiments

**Datasets and Evaluation Metrics**: Following [23, 12, 45, 43, 5], we use 800 high-quality (2K resolution) images from DIV2K dataset [34] as training set. We evaluate our models on five standard benchmarks: Set5 [1], Set14 [41], BSD100 [26], Urban100 [14] and Manga109 [27] in three upscaling factors: $\times 2$, $\times 3$ and $\times 4$. All the experiments are conducted with Bicubic (BI) downsampling degradation. The estimated high-resolution images are evaluated by PSNR and SSIM [39] on Y channel (i.e., luminance) of the transformed YCbCr space.

**Training Settings**: We crop the HR patches from DIV2K dataset [34] for training. Then these patches are downsampled by Bicubic to get the LR patches. For all different downsampling scales in our experiments, we fixed the size of LR patches as $60 \times 60$. All the training patches are augmented by randomly horizontally flipping and ratation of $90°$, $180°$, $270°$ [23]. We set the minibatch size to 4 and train our model using ADAM [18] optimizer with the settings of $\beta_1 = 0.9$, $\beta_2 = 0.999$, $\epsilon = 10^{-8}$. The initial learning rate is set as $10^{-4}$ and then decreases to half for every $2 \times 10^5$ iterations. Training is terminated after $8 \times 10^5$ iterations. The network is trained by using $\ell_1$ norm loss. The IGNN is implemented on the PyTorch framework on an NVIDIA Tesla V100 GPU.

In the Graph Aggregation module, we set query patch size $l$ as $3 \times 3$ and the number of neighbors $k$ as 5. The size of the searching window $d$ is 30 within the $s$ times downsampled LR (i.e., $E_{L\downarrow s}$). Note that our GraphAgg is a plug-in module, and the backbone of our network is based on EDSR. We use the pretrained backbone model to initialize the IGNN in order to improve the training stability and save the training time.

**Table 1:** Quantitative results in comparison with the state-of-the-art methods. Average PSNR/SSIM for scale factor $\times 2$, $\times 3$ and $\times 4$ on benchmark datasets Set5, Set14, BSD100, Urban100, and Manga109. Best and second best performance are **highlighted** and underlined.

| Method | Scale | Set5 PSNR | Set5 SSIM | Set14 PSNR | Set14 SSIM | BSD100 PSNR | BSD100 SSIM | Urban100 PSNR | Urban100 SSIM | Manga109 PSNR | Manga109 SSIM |
|---|---|---|---|---|---|---|---|---|---|---|---|
| VDSR [16] | $\times 2$ | 37.53 | 0.9590 | 33.05 | 0.9130 | 31.90 | 0.8960 | 30.77 | 0.9140 | 37.22 | 0.9750 |
| LapSRN [20] | $\times 2$ | 37.52 | 0.9591 | 33.08 | 0.9130 | 31.08 | 0.8950 | 30.41 | 0.9101 | 37.27 | 0.9740 |
| MemNet [33] | $\times 2$ | 37.78 | 0.9597 | 33.28 | 0.9142 | 32.08 | 0.8978 | 31.31 | 0.9195 | 37.72 | 0.9740 |
| DBPN [12] | $\times 2$ | 38.09 | 0.9600 | 33.85 | 0.9190 | 32.27 | 0.9000 | 32.55 | 0.9324 | 38.89 | 0.9775 |
| RDN [45] | $\times 2$ | 38.24 | 0.9614 | 34.01 | 0.9212 | 32.34 | 0.9017 | 32.89 | 0.9353 | 39.18 | 0.9780 |
| NLRN [24] | $\times 2$ | 38.00 | 0.9603 | 33.46 | 0.9159 | 32.19 | 0.8992 | 31.81 | 0.9249 | – | – |
| RNAN [44] | $\times 2$ | 38.17 | 0.9611 | 33.87 | 0.9207 | 32.32 | 0.9014 | 32.73 | 0.9340 | 39.23 | 0.9785 |
| SRFBN [22] | $\times 2$ | 38.11 | 0.9609 | 33.82 | 0.9196 | 32.29 | 0.9010 | 32.62 | 0.9328 | 39.08 | 0.9779 |
| OISR-RK3 [13] | $\times 2$ | 38.21 | 0.9612 | 33.94 | 0.9206 | 32.36 | 0.9019 | 33.03 | 0.9365 | 39.20 | 0.9782 |
| SAN [5] | $\times 2$ | **38.31** | **0.9620** | 34.07 | 0.9213 | 32.42 | 0.9028 | 33.10 | 0.9370 | 39.32 | **0.9792** |
| EDSR [23] | $\times 2$ | 38.11 | 0.9602 | 33.92 | 0.9195 | 32.32 | 0.9013 | 32.93 | 0.9351 | 39.10 | 0.9773 |
| IGNN (Ours) | $\times 2$ | 38.24 | 0.9613 | 34.07 | 0.9217 | 32.41 | 0.9025 | 33.23 | 0.9383 | 39.35 | 0.9786 |
| IGNN+ (Ours) | $\times 2$ | **38.31** | 0.9616 | 34.18 | 0.9222 | 32.46 | 0.9030 | 33.42 | 0.9396 | 39.54 | 0.9790 |
| VDSR [16] | $\times 3$ | 33.67 | 0.9210 | 29.78 | 0.8320 | 28.83 | 0.7990 | 27.14 | 0.8290 | 32.01 | 0.9340 |
| LapSRN [20] | $\times 3$ | 33.82 | 0.9227 | 29.87 | 0.8320 | 28.82 | 0.7980 | 27.07 | 0.8280 | 32.21 | 0.9350 |
| MemNet [33] | $\times 3$ | 34.09 | 0.9248 | 30.00 | 0.8350 | 28.96 | 0.8001 | 27.56 | 0.8376 | 32.51 | 0.9369 |
| RDN [45] | $\times 3$ | 34.71 | 0.9296 | 30.57 | 0.8468 | 29.26 | 0.8093 | 28.80 | 0.8653 | 34.13 | 0.9484 |
| NLRN [24] | $\times 3$ | 34.27 | 0.9266 | 30.16 | 0.8374 | 29.06 | 0.8026 | 27.93 | 0.8453 | - | - |
| RNAN [44] | $\times 3$ | 34.66 | 0.9290 | 30.52 | 0.8463 | 29.26 | 0.8090 | 28.75 | 0.8646 | 34.25 | 0.9483 |
| SRFBN [22] | $\times 3$ | 34.70 | 0.9292 | 30.51 | 0.8461 | 29.24 | 0.8084 | 28.73 | 0.8641 | 34.18 | 0.9481 |
| OISR-RK3 [13] | $\times 3$ | 34.72 | 0.9297 | 30.57 | 0.8470 | 29.29 | 0.8103 | 28.95 | 0.8680 | 34.32 | 0.9493 |
| SAN [5] | $\times 3$ | 34.75 | 0.9300 | 30.59 | 0.8476 | 29.33 | 0.8112 | 28.93 | 0.8671 | 34.30 | 0.9494 |
| EDSR [23] | $\times 3$ | 34.65 | 0.9280 | 30.52 | 0.8462 | 29.25 | 0.8093 | 28.80 | 0.8653 | 34.17 | 0.9476 |
| IGNN (Ours) | $\times 3$ | 34.72 | 0.9298 | 30.66 | 0.8484 | 29.31 | 0.8105 | 29.03 | 0.8696 | 34.39 | 0.9496 |
| IGNN+ (Ours) | $\times 3$ | **34.84** | **0.9305** | **30.75** | **0.8496** | **29.37** | **0.8115** | **29.20** | **0.8721** | **34.67** | **0.9509** |
| VDSR [16] | $\times 4$ | 31.35 | 0.8830 | 28.02 | 0.7680 | 27.29 | 0.0726 | 25.18 | 0.7540 | 28.83 | 0.8870 |
| LapSRN [20] | $\times 4$ | 31.54 | 0.8850 | 28.19 | 0.7720 | 27.32 | 0.7270 | 25.21 | 0.7560 | 29.09 | 0.8900 |
| MemNet [33] | $\times 4$ | 31.74 | 0.8893 | 28.26 | 0.7723 | 27.40 | 0.7281 | 25.50 | 0.7630 | 29.42 | 0.8942 |
| DBPN [12] | $\times 4$ | 32.47 | 0.8980 | 28.82 | 0.7860 | 27.72 | 0.7400 | 26.38 | 0.7946 | 30.91 | 0.9137 |
| RDN [45] | $\times 4$ | 32.47 | 0.8990 | 28.81 | 0.7871 | 27.72 | 0.7419 | 26.61 | 0.8028 | 31.00 | 0.9151 |
| NLRN [24] | $\times 4$ | 31.92 | 0.8916 | 28.36 | 0.7745 | 27.48 | 0.7306 | 25.79 | 0.7729 | - | - |
| RNAN [44] | $\times 4$ | 32.49 | 0.8982 | 28.83 | 0.7878 | 27.72 | 0.7421 | 26.61 | 0.8023 | 31.09 | 0.9149 |
| SRFBN [22] | $\times 4$ | 32.47 | 0.8983 | 28.81 | 0.7868 | 27.72 | 0.7409 | 26.60 | 0.8015 | 31.15 | 0.9160 |
| OISR-RK3 [13] | $\times 4$ | 32.53 | 0.8992 | 28.86 | 0.7878 | 27.75 | 0.7428 | 26.79 | 0.8068 | 31.26 | 0.9170 |
| SAN [5] | $\times 4$ | 32.64 | 0.9003 | 28.92 | 0.7888 | 27.78 | 0.7436 | 26.79 | 0.8068 | 31.18 | 0.9169 |
| EDSR [23] | $\times 4$ | 32.46 | 0.8968 | 28.80 | 0.7876 | 27.71 | 0.7420 | 26.64 | 0.8033 | 31.02 | 0.9148 |
| IGNN (Ours) | $\times 4$ | 32.57 | 0.8998 | 28.85 | 0.7891 | 27.77 | 0.7434 | 26.84 | 0.8090 | 31.28 | 0.9182 |
| IGNN+ (Ours) | $\times 4$ | **32.71** | **0.9011** | **28.96** | **0.7908** | **27.84** | **0.7447** | **27.04** | **0.8128** | **31.59** | **0.9207** |

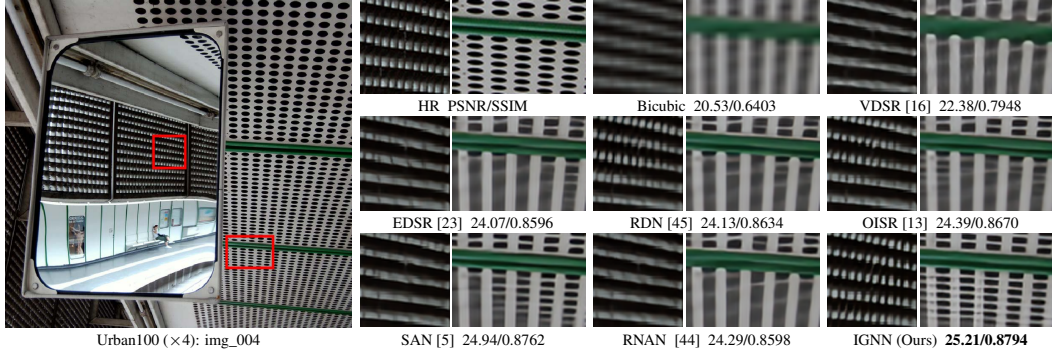

**Figure 3:** Visual results with Bicubic downsampling (×4) on "img_004" from Urban100. The proposed method recovers more details.

## 3.1 Comparisons with State-of-the-Art Methods

We compare our proposed method with 11 state-of-the-art methods: VDSR [16], LapSRN [20], MemNet [33], EDSR [23], DBPN [12], RDN [45], NLRN [24], RNAN [44], SRFBN [22], OISR [13], and SAN [5]. Following [23, 35, 44, 5], we also use self-ensemble strategy to further improve our IGNN and denote the self-ensembled one as IGNN+.

As shown in Table 1, the proposed IGNN outperforms existing CNN-based methods, e.g. VDSR [16], LapSRN [20], MemNet [33], EDSR [23], DBPN [12], RDN [45], SRFBN [22] and OISR [13], and existing non-local neural networks, e.g. NLRN [24] and RNAN [44]. Similar to OISR [13], IGNN is also built based on EDSR [23] but has better performance. This demonstrates the effectiveness of the proposed GraphAgg for SISR. In addition, the GraphAgg only has two very small sub-networks (ECN and DEN), each of which only contain three convolutional layers. Thus, the improvement comes from the cross-scale aggregation rather than a larger model size. As to SAN [5], it performs the best in some cases. However, it uses a very deep network (including 200 residual blocks) which is around *seven times deeper* than the proposed IGNN.

We also present a qualitative comparison of our IGNN with other state-of-the-art methods, as shown in Figure 3. The IGNN recovers more details with less blurring, especially on small recurring textures. This results demonstrate that IGNN indeed explores the rich texture from cross-scale patch searching and aggregation. Compared with other methods, IGNN obtains image-specific information from the searched $k$ HR feature patches. Such internal cues complement external information obtained by network learning from the dataset. More visual results are provided in the supplementary material.

## 3.2 Analysis and Discussions

In this section, we conduct a number of comparative experiments for further analysis and discussions.

**Effectiveness of Graph Aggregation Module:** In order to show the effectiveness of the cross-scale aggregation intuitively, we provide a non-learning version, denoted as GraphAgg*, which constructs the cross-scale graph in exactly the same way as to IGNN. Different from IGNNwhose GraphAgg aggregates extracted features in IGNN, GraphAgg* directly aggregates $k$ neighboring HR patches cropped from the input LR image by simply averaging. As shown in the first row of Figure 4, GraphAgg* is capable of recovering more detailed and sharper result, compared with the Bicubic upsampled input LR image. The results intuitively show the effectiveness of cross-scale aggregation in image SR task. Even though the SR images generated from GraphAgg* are promising, they still contain some artifacts in the second row of Figure 4. The proposed IGNN can remove them and restore better images with finer details by aggregating the features extracted from the network.

To further verify the effectiveness of GraphAgg that aggregation information across different scales, we replace it with the basic non-local block [37, 24] (fully-connected aggregation) and KNN ($k$ neighbors aggregation) within the same scale. The results in Table 2 show that the basic non-local blocks bring limited improvements of only 0.05 dB in PSNR. Besides, our GraphAgg (cross-scale) outperforms KNN (same-scale) by a considerable margin. In contrast, IGNN shows evident improvements in performance, suggesting the importance of cross-scale aggregation for SISR.

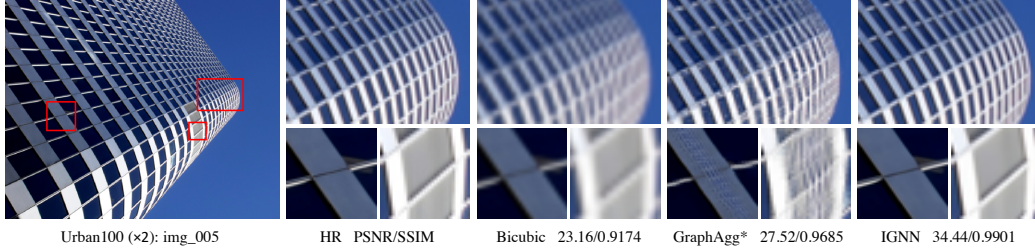

| Urban100 (×2): img_005 | HR PSNR/SSIM | Bicubic 23.16/0.9174 | GraphAgg* 27.52/0.9685 | IGNN 34.44/0.9901 |

**Figure 4:** Visual results with Bicubic downsampling (×2) on "img_005" from Urban100. The GraphAgg* denotes the non-learning version of GraphAgg, which aggregates cross-scale information in the image domain. Compared with the Bicubic upsampled result, GraphAgg* recovers sharper details, suggesting the effectiveness of the proposed cross-scale aggregation. However, GraphAgg* still generates some artifacts with a direct aggregation (*patch2img*) in image domain. By aggregating in feature domain, the proposed IGNN removes the artifacts and generates better SR image.

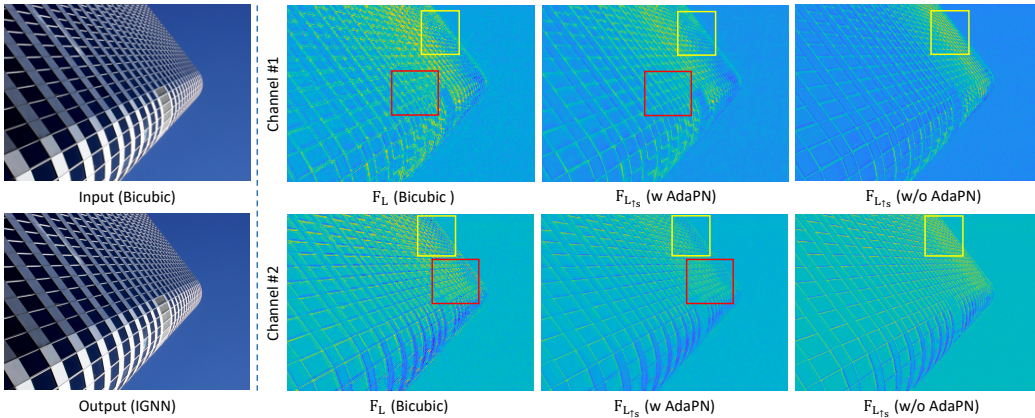

**Figure 5:** Feature visualization for validation the effectiveness of the GraphAgg and AdaPN. The two rows show the different feature maps of two channels. $F_L$ (Bicubic) represents LR features $F_L$ with Bicubic upsampling. $F_{L\uparrow s}$ (w AdaPN) is denoted as aggregated HR features by GraphAgg, and $F_{L\uparrow s}$ (w/o AdaPN) represents HR features without normalization by AdaPN. As shown in the red boxes, $F_{L\uparrow s}$ (w AdaPN) contain more rich and sharp details than $F_L$ in both two channels. Compared with $F_{L\uparrow s}$ (w/o AdaPN), $F_{L\uparrow s}$ (w AdaPN) maintain better consistency with $F_L$ in low-frequency discrepancy (e.g., color), as shown in the yellow boxes.

**Table 2:** Comparison GraphAgg with the Non-local block on Urban100 (×2).

|  | Baseline | Non-local | KNN (same-scale) | IGNN |
|---|---|---|---|---|
| PSNR | 32.93 | 32.98 | 33.01 | **33.23** |
| SSIM | 0.9351 | 0.9362 | 0.9364 | **0.9383** |

**Table 3:** Results on Urban100 (×2) for different positions of GraphAgg in the network.

|  | after 8th | after 16th | after 24th |
|---|---|---|---|
| PSNR | 33.17 | **33.23** | 33.19 |
| SSIM | 0.9378 | **0.9383** | 0.9380 |

**Feature Visualization:** To validate the effectiveness of the GraphAgg and AdaPN, we also present the intermediate features from two channels in Figure 5. According to the comparison in red boxes between LR features $F_L$ (Bicubic) with Bicubic upsampling and HR features $F_{L\uparrow s}$ (w AdaPN), It is obvious that $F_{L\uparrow s}$ contains more rich and sharp details, suggesting the effectiveness of GraphAgg in obtaining more detailed textures in the feature domain.

From the yellow boxes in the three features (in the same row) shown in Figure 5, HR features $F_{L\uparrow s}$ (w/o AdaPN) without Adaptive Patch Normalization exhibit some discrepancy in low-frequency information (e.g., color) with input LR features $F_L$ (Bicubic). However, this discrepancy is almost eliminated in HR features $F_{L\uparrow s}$ (w AdaPN) with normalization by AdaPN. It shows that AdaPN makes the GraphAgg module more accurate and robust in patch aggregation.

**Position of Graph Aggregation Module:** We compare three positions in the backbone network to integrate GraphAgg, i.e., after the 8th residual block, after the 16th residual block and after the 24th residual block. As summarized in Table 3, performance improvement is observed at all positions. The largest gain is achieved by inserting GraphAgg in the middle, i.e., after the 16th residual block.

**Table 4:** Results on Urban100 ($\times 2$) for varying sizes $d$ of searching window.

|  | $d=10$ | $d=20$ | $d=30$ | $d=LR_{\downarrow s}$ |
|---|---|---|---|---|
| PSNR | 33.14 | 33.18 | 33.23 | **33.24** |
| SSIM | 0.9372 | 0.9380 | **0.9383** | **0.9383** |

**Table 5:** Results on Urban100 ($\times 2$) for varying neighbor number $k$.

|  | $k=1$ | $k=3$ | $k=5$ | $k=7$ | $k=9$ | $k=11$ |
|---|---|---|---|---|---|---|
| PSNR | 33.17 | 33.21 | **33.23** | **33.23** | 33.22 | 33.22 |
| SSIM | 0.9377 | 0.9381 | **0.9383** | 0.9382 | 0.9382 | 0.9381 |

**Table 6:** Results on Urban100 ($\times 2$) for variants of GraphAgg. The (w/o AdaPN), (w/o ECN), and (w/o AdaPN, w/o ECN) denote removing Adaptive Patch Normalization, removing Edge-Conditioned sub-network, and removing both of them, respectively.

|  | w/o AdaPN | w/o ECN | w/o AdaPN and w/o ECN | IGNN |
|---|---|---|---|---|
| PSNR | 33.18 | 33.12 | 33.09 | **33.23** |
| SSIM | 0.9379 | 0.9372 | 0.9369 | **0.9383** |

**Settings for Graph Aggregation Module:** We investigate the influence of the searching window size $d$ and neighborhood number $k$ in GraphAgg. Table 4 shows the results on Urban100 ($\times 2$) for different size of searching window $d$. As expected, the estimated SR image has better quality when $d$ increases. We also find that $d=30$ has almost the same performance relative to searching among the whole downsampled features $E_{L\downarrow s}$ ($d=LR_{\downarrow s}$). Therefore, we empirically set $d=30$ (i.e., $30 \times 30$ window) as a trade-off between the computational complexity and performance.

Table 5 presents the results on Urban100 ($\times 2$) for different number of neighbors $k$. In general, more neighbors improve SR results since more HR information can be utilized by GraphAgg. However, the performance does not improve after $k=5$ since it may be hard to find more than five useful HR neighbors for aggregation.

**Effectiveness of Adaptive Patch Normalization and Edge-Conditioned sub-network:** The retrieved $k$ HR neighboring patches are sometimes mismatched with the query patch in low-frequency information, e.g., color, brightness. To solve this problem, we adopt two modules in the proposed GraphAgg, i.e., Adaptive Patch Normalization (AdaPN) and Edge-Conditioned sub-network (ECN), i.e., $\mathcal{F}_\theta\left(\mathcal{D}^{n_r \to q}\right)$. To validate the effectiveness of AdaPN and ECN, we compare GraphAgg with three variants: removing AdaPN only (w/o AdaPN), removing ECN only (w/o ECN), and removing both of them (w/o AdaPN and w/o ECN). Table 6 shows that the network performs worse when any one of them is removed. Note that we remove ECN by replacing $\exp\left(\mathcal{F}_\theta\left(\mathcal{D}^{n_r \to q}\right)\right)$ in Eq. (2) by the metric of weighted Euclidean distance with Gaussian kernel, i.e., $\exp\left(-\|\mathcal{D}^{n_r \to q}\|_2^2/10\right)$. The above experimental results demonstrate that AdaPN and ECW indeed make the GraphAgg module more robust for the patch aggregation.

# 4 Conclusion

We present a novel notion of modelling internal correlations of cross-scale recurring patches as a graph, and then propose a graph network IGNN that explores this internal recurrence property effectively. IGNN obtains rich textures from the $k$ HR counterparts found from LR features itself to alleviate the ill-posed nature in SISR and recover more detailed textures. The paper has shown the effectiveness of the cross-scale graph aggregation, which passes HR information from HR neighboring patches to LR ones. Extensive results over benchmarks demonstrate the effectiveness of the proposed IGNN against state-of-the-art SISR methods.

# Acknowledgments

This research is conducted in collaboration with SenseTime and supported by the Singapore Government through the Industry Alignment Fund-Industry Collaboration Projects Grant. It is also partially supported by Singapore MOE AcRF Tier 1 (2018-T1-002-056) and NTU SUG.

## Broader Impact

This paper is an exploratory work on single image super-resolution using graph convolutional network. The main impacts of this work are academia-oriented, it is expected to promote the research progress of image super-resolution and motivate novel methods in the related fields.

As for future societal influence, this work will largely improve the quality of pictures taken by cameras or other mobile devices such as smartphones. In addition, it enhances public safety monitored by vision systems via the higher resolution of the monitoring view. Though there might be some potential risks, e.g., criminals might use this technology for peeping into peoples' personal privacy. It is worth noticing that the positive social effects of this technology far exceeds the potential problems. We call on people to use this technology and its derivative applications without compromising the personal interest of the general public.

## Footnotes

[1] Two $\times 2$ upsample *PixelShuffle* operations can achieve $\times 4$ upsampling [23].

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
