[Supplementary Material]

# Cross-Scale Internal Graph Neural Network for Image Super-Resolution

## (Supplementary Materials)

**Shangchen Zhou**[1]    **Jiawei Zhang**[2]    **Wangmeng Zuo**[3]    **Chen Change Loy**[1]*
[1]Nanyang Technological University    [2]SenseTime Research    [3]Harbin Institute of Technology
{s200094,ccloy}@ntu.edu.sg   zhangjiawei@sensetime.com   wmzuo@hit.edu.cn
https://github.com/sczhou/IGNN

In this supplementary material, we provide additional details and results to the paper. In Sec. A, we first present the detailed architectures of two small sub-networks in the proposed Graph Aggregation module (GraphAgg). Then, we give an illustration of operation details in the GraphAgg. Sec. B presents further analysis and discussions on our proposed GraphAgg module and IGNN network. Finally, we show more visual experimental results compared with other state-of-the-art SR networks in Sec. C.

## A  Details in GraphAgg

### A.1  Architecture Details

As presented in Sec. 2.2 in the manuscript, the proposed GraphAgg has two small sub-networks: Edge-Conditioned sub-network (ECN) and Downsampled-Embedding sub-network (DEN). Tables 1 and 2 list the detailed configurations of ECN and DEN, respectively. In Graph Construction, we use the first three layers of the VGG19 [5] with fixed pre-trained parameters.

**Table 1:** Architecture of the Edge-Conditioned sub-network (ECN).

| Layer | Kernel | Stride | Padding | Feature |
|---|---|---|---|---|
| Input ($\mathcal{D}^{n_r \to q}$) | | | | 64 |
| Conv/ReLU | $1 \times 1$ | 1 | 1 | 64 |
| Conv/ReLU | $1 \times 1$ | 1 | 1 | 64 |
| Conv | $1 \times 1$ | 1 | 1 | 1 |

**Table 2:** Architecture of the Downsampled-Embedding sub-network (DEN).

| Layer | Kernel | Stride | Padding | Feature |
|---|---|---|---|---|
| Input ($F_{L\uparrow s}$) | | | | 256 |
| Conv/ReLU | $5 \times 5$ | 1 | 1 | 256 |
| Conv/ReLU | $3 \times 3$ | 1 | 1 | 256 |
| Conv | $3 \times 3$ | 1 | 1 | 256 |

### A.2  Illustration of Detailed Processes in GraphAgg

To further clarify the operations in the GraphAgg, we illustrate the details as shown in Figure 1.

In Figure 1(a), we extract $l \times l$ LR patches using *img2patch* operation with a stride of $g$ from features $E_{L\downarrow s}$ and $E_L$, where we set $l = 3$ and $g = 2$ in our network. Thus we obtain $m_1 \times n_1$ LR patches (denoted as $\mathcal{V}_1^l$) and $m_2 \times n_2$ LR patches (denoted as $\mathcal{V}_2^l$) from $E_{L\downarrow s}$ and $E_L$ respectively. Denote the feature shapes of $E_{L\downarrow s}$ and $E_L$ as $H/s \times W/s$ and $H \times W$ respectively. Therefore, $m_1 = \lfloor (H/s-l)/g \rfloor + 1, n_1 = \lfloor (W/s-l)/g \rfloor + 1$, and $m_2 = \lfloor (H-l)/g \rfloor + 1, n_2 = \lfloor (W-l)/g \rfloor + 1$. Each LR patch in $\mathcal{V}_2^l$ find the $k$ nearest neighboring LR patches from $\mathcal{V}_1^l$ according to the Euclidean distance. Note that we do not consider the searching window in this discussion for simplicity.

To obtain $k$ corresponding HR ($ls \times ls$) patch regions in $E_L$ scale, we map each LR patch in $E_{L\downarrow s}$ to HR regions in $E_L$ scale, as shown in Figure 1(b). In $E_L$ scale, the $ls \times ls$ patch regions are obtained

---

(a) Find $k$ nearest neighboring patches in Graph Construction.

(b) Vertex Mapping in Graph Construction.

(c) Patch2Img in Patch Aggregation.

**Figure 1:** In (a), $n_1 \times m_1$ LR patches and $n_2 \times m_2$ LR patches are extracted using *img2patch* operation with a stride of $g$ from features $E_{L\downarrow s}$ and $E_L$, respectively. Each LR patch of $E_L$ find the $k$ nearest neighboring LR patches from $E_{L\downarrow s}$. In (b), each found LR patch in $E_{L\downarrow s}$ is mapped to a HR region in $E_L$ scale. In $E_L$ scale, the HR patch regions are extracted using *img2patch* operation with the stride of $gs$. In (c), we transform $n_2 \times m_2$ HR patches to the final HR feature $F_{L\uparrow s}$ using a *patch2img* operation with the stride of $gs$. Note that the sizes of LR and HR patches are $l \times l$ and $ls \times ls$, respectively, where $s$ is desired upsampling factor. Refer to Sec. A.2 for more details.

using *img2patch* operation with the stride of $gs$. It exactly has the same number ($m_1 \times n_1$) of patches as LR patches in $E_{L\downarrow s}$, i.e., $\lfloor (H - ls)/gs \rfloor + 1 = m_1$, $\lfloor (W - ls)/gs \rfloor + 1 = n_1$. Therefore, the LR and HR patch regions from $E_{L\downarrow s}$ and $E_L$ scales can be matched one-on-one.

As presented in Eq. (2) in the manuscript, we obtain one aggregated HR patch for each LR patch in the LR patch set $\mathcal{V}_2^l$, which contains $m_2 \times n_2$ LR patches. Thus, we obtain a HR patch set $\mathcal{V}^{rs}$ containing $m_2 \times n_2$ HR patches with $ls \times ls$ patch size. Figure 1(c) shows that we take the $\mathcal{V}^{rs}$ as input and use a *patch2img* operation with the stride of $gs$ to generate the HR features $F_{L\uparrow s}$.

## B  More Discussions on GraphAgg and IGNN

In this section, we first present more ablation experiments to demonstrate the effectiveness of the proposed IGNN further, including the effect of using $F'_L$ and $F_{L\uparrow s}$ and number of GraphAgg modules inserted in networks. In addition, we report and compare the runtime of the state-of-the-art networks and the proposed IGNN.

### B.1  Effectiveness of $F'_L$ and $F_{L\uparrow s}$

To validate the effectiveness of both enriched features $F'_L$ and aggregated HR features $F_{L\uparrow s}$, we compare our network with three variant networks: replacing the enriched LR features $F'_L$ by the original LR features $F_L$ (w/o $F'_L$), removing aggregated HR features $F_{L\uparrow s}$ (w/o $F_{L\uparrow s}$) and without both of them (baseline), i.e., EDSR. According to Table 3, these three variant networks generate worse SR results compared to the completed network.

### B.2  Effectiveness of of Multiple GraphAgg Module

To explore whether the number of GraphAgg module affects IGNN performance, we evaluate to insert 1 (after 16th residual block), 2 (after 8th and 24th residual block), and 3 (after 8th, 16th, and 24th residual block) GraphAgg modules in the backbone network, respectively. As shown in Table 4, using more GraphAgg module only leads to slight PSNR/SSIM gains. Thus we only employ one

**Table 3:** Results on Urban100 ($\times 2$) for different variants of networks. The (w/o $F_L'$) represents replacing the enriched LR features $F_L'$ by the original LR features $F_L$, and (w/o $F_{L\uparrow s}$) represents removing the aggregated HR features $F_{L\uparrow s}$. Note the EDSR is our baseline which is equivalent to removing both $F_L'$ and $F_{L\uparrow s}$.

|  | baseline (EDSR) | w/o $F_L'$ | w/o $F_{L\uparrow s}$ | IGNN |
|---|---|---|---|---|
| PSNR | 32.93 | 33.19 | 33.13 | **33.23** |
| SSIM | 0.9351 | 0.9379 | 0.9374 | **0.9383** |

GraphAgg module in our IGNN as a trade-off among the computational complexity and performance. Compared with the baseline network (EDSR) without GraphAgg inserted, our proposed IGNN shows a large performance gain.

**Table 4:** Results on Urban100 ($\times 2$) for different numbers of GraphAgg modules are inserted in the networks.

|  | 0 (baseline) | **1 (after 16th)** | 2 (after 8th and 24th ) | 3 (after 8th, 16th, and 24th ) |
|---|---|---|---|---|
| PSNR | 32.93 | 33.23 | 33.23 | 33.25 |
| SSIM | 0.9351 | 0.9383 | 0.9384 | 0.9385 |

## B.3 Relationship between Performance Gain and Self-similarity Level

It is worth to analyze further on performance gains for different self-similarity levels. As shown in Figure 2, our method performs better in regions with self-similarity, especially in regions where texture patterns are extremely small. Besides, the performance can also be well maintained to that of EDSR in regions with few self-similar patches.

Self-similarity Level          PSNR Gain

**Figure 2:** Examples to show the relationship between self-similarity level and PSNR gain (over EDSR). The brighter regions indicate larger values.

## B.4 Runtime

Here, we report and compare the runtime of state-of-the-art networks [8, 4, 7, 2, 1] and the proposed IGNN. All existing methods are evaluated using their publicly available code. As shown in Table 5, the proposed network has comparable runtime as [8, 4, 7, 2], but it has better performance on all benchmarks at all scales (Refer to Table 1 in the manuscript). As for SAN [1], the proposed IGNN runs over two times faster than it, and performs better in most cases.

**Table 5:** Runtime of different networks. All methods are evaluated on an NVIDIA Tesla V100 GPU.

|  | RDN [8] | EDSR [4] | RNAN [7] | OISR-RK3 [2] | SAN [1] | IGNN (Ours) |
|---|---|---|---|---|---|---|
| PSNR | 32.89 | 32.93 | 32.73 | 33.03 | 33.10 | **33.23** |
| Time (sec) | 1.538 | 1.416 | 2.280 | 1.833 | 5.971 | 2.676 |

# C  Qualitative Comparisons

In this section, we provide more visual comparisons with seven state-of-the-art SISR networks, i.e., VDSR [3], EDSR [4], RDN [8], RCAN [6], OISR [2], SAN [1], and RNAN [7], on standard benchmark datasets. As shown in Figure 3 and Figure 4, the proposed IGNN recovers richer and sharper details from the LR images especially in the regions with recurring patterns.

**Figure 3:** Visual comparison for ×4 SR on benchmark datasets.

**Figure 4:** Visual comparison for ×4 SR on benchmark datasets.