[Reviews · NeurIPS 2020]

Review 1

Summary and Contributions: This paper proposes adding a graph convolution layer to the EDSR superresolution network. This layer combines the graph convolution with the idea of internal self-similarity across scales, effectively extracting higher resolution patches that would match a query patch after downsampling. Quantitative metric comparisons and qualitative image patch comparisons are provided, showing that the proposed module improves performance in images with high self-similarity.

Strengths: Taking advantage of non-local self-similarity by explicitly encoding this into a network layer is a good idea (though this has been used before in some of the cited papers, like [23, 28, 34]). The image examples shown demonstrate that the addition of the graph convolution module does improve performance over the EDSR baseline in regions of repeating texture.

Weaknesses: As in many SISR papers these days, the metric increase in PSNR and SSIM is vanishingly small. The qualitative example shown in Fig 3 demonstrates a more significant boost in metrics, however, this is obviously not reflected in the datasets as a whole given the average values reported in Table 1. I think it would be valid to state that this method mostly only helps in regions of self-similarity; it would be interesting to see if there is a metric that could capture this though rather than relying on anecdotal crops. The difference from previous graph convolutions applied for image restoration (like in [34]) seems fairly minor. Even the module as a whole is effectively a pretty small change to EDSR, involving very few additional layers. It is unclear whether the idea of finding self-similar patches from the downsampled LR image is helping, I did not see an ablation demonstrating the performance when finding the patches based on E_{L \down s} versus just taking them from the same scale E_L, which would help bolster the claim that the higher resolution matches are helping. The introduction claims (L45-49) that patches from the same scale with subpixel misalignment would “hardly improve” performance, but there are entire papers written claiming the opposite, such as: “Handheld Multi-Frame Super-Resolution”, Wronski et al 2019 The paper is hard to read. The figures are not bad, but the explanatory text in Section 2 is bogged down by a ton of notation: \mathcal G, V, E, D, F; regular I, E, F; subscripts/superscripts L, l, ls, L \down s, L \up s, n_r, q, \theta, c and names/acronyms: IGCN, GraphAgg, ECN, DEN, AdaPN This really obscures a not-so-complicated idea in a confusing way. Using bicubic downsampling for factors > 2 seems like bad signal processing practice (I acknowledge that this is a flaw that most synthetic SISR papers have and is not really the authors’ fault). In addition, I wonder whether the benefit of this method is dependent on the downsampling method used to get the graph connections matching the one used for mapping HR ground truth to LR input. The table reports results on IGCN and ensemble IGCN+ but only includes EDSR and not the ensemble EDSR+, which seems unfair.

Correctness: As far as I can tell, everything is correct.

Clarity: As explained in the “Weaknesses” section, I found this paper hard to read.

Relation to Prior Work: One missing citation is “Zero-Shot Super-Resolution using Deep Internal Learning”, Shocher et al 2017. It is a bit hard to tell exactly how this version of graph convolution differs from [34] besides pulling the patches from a different scale.

Reproducibility: Yes

Additional Feedback: Post-rebuttal feedback: After reading the other reviews and rebuttal, I have increased my score. I appreciate the thorough rebuttal, which addresses whether the cross-scale aggregation provides benefit (Table 1) and shows further analysis into relative improvement for different components of the image (Figure 1), which I think would make valuable additions to the revised text.


Review 2

Summary and Contributions: This paper presents a method for image super-resolution, by exploiting the similarity of patches across different scales through cross-scale graph convolutional network. While using the similar patches for this problem has been explored many times previously, the paper is novel in the sense of cross-scale similarity and using graph representation to solve the problem.

Strengths: Non-local self-similarity is one of the keys in solving the image restoration problems including the super-resolution problem. The authors push more into direction by exploiting multi scales for the searcch, and to the best of my knowledge, this is a novel and good idea. The way the authors frame the solution through graph convolution was also interesting. The proposed algorithm also outperforms previous works.

Weaknesses: While the performance of the proposed algorithm is better than previous works, I am not sure if the difference is significant. Personally, I think that SR research needs to focus more on perceptual quality as increasing PSNR does not guarantee good image quality. It would be interesting to see if the proposed algorithm can be coupled with perceptual GAN-based methods (use it as a generator). Another important direction for SR is the efficiency. I did not see any discussion on this issue. How the the computation compare to previous work? What is the runtime? Overall, satisfied with the rebuttal and I am maintaining my original score.

Correctness: The paper seems to be correct.

Clarity: Yes, the paper is well written.

Relation to Prior Work: Yes.

Reproducibility: Yes

Additional Feedback: N/A.


Review 3

Summary and Contributions: This paper proposes a graph convolutional network based super-resolution framework that leverages abundant external data as well as internal image statistics by using the proposed cross-scale graph aggregation module that works in the feature space. The graph aggregation module searches for k similar patches in the lower scale of the embedded space, constructs a graph, and aggregates the patches using weights generated by the edge-conditioned subnet. The aggregated patches (in the feature domain) are concatenated in the middle of the CNN, as well as before the last convolution layer. This paper has many contributions: (i) a novel cross-scale graph-based graph aggregation module that uses internal information, (ii) a novel SR framework that uses external as well as internal images, (iii) extensive analysis on the proposed framework, (iv) state-of-the-art performance showing the potential of this approach on widely used SR benchmarks.

Strengths: - A novel approach using both internal image distributions as well as external data for training. - Very well-designed graph aggregation module for constructing graphs and aggregating patches for leveraging internal patches. - Extensive experiments (in the main paper as well as the supplementary material) that help analyze the framework and gain insights into where the benefits are. - State-of-the-art performance on common SR benchmarks. This is a great paper with an intuitive and novel framework, backed up by extensive analysis and state-of-the-art performance. I expect this paper to add significant contributions to the community.

Weaknesses: - What happens if similar patches are inexistent in the down-scaled image? (e.g. if the image contains small single objects?) If k=5 patches cannot be found, more error may be caused by the aggregating these patches. What is expected of the ECN output (weight value) for this patch? I reckon the weight cannot be 0 due to the exponential term. - I think the optimal value of d and k may be different depending on the input resolution, hence, the framework may be sub-optimal for some input resolutions. - An inherent weakness of using a non-local type approach would be the processing time. But this paper overcomes this well by k-neighbor searches in a pre-defined window. It is faster than the recent state-of-the art SR method SAN.

Correctness: The claims and the method are sound. The empirical methodology is also correct.

Clarity: The paper is clearly written.

Relation to Prior Work: Overall, it is clearly discussed how this method differs from prior work. However, it could help to mention the difference to previous exemplar-based methods [9, 43, 27].

Reproducibility: Yes

Additional Feedback: Suggestions: - It would help the readers to gain a better understanding of the context to mention the difference to previous exemplar-based methods [9, 43, 27]. - I suggest the authors to release the code, which can be crucial in the paper’s reproducibility, especially if it contains complex modules. Comments: - The benchmark results make sense that this framework performs better for large resolution benchmarks (Urban100, Manga109) since they are more likely to contain more internal recurrence. (especially Urban100, since it contains many building objects in images) I think this observation could be worth mentioning in the paper. Minor: - Reference [41] incomplete - p.4 line 124 yellow square -> blue square ==================================== After having read the rebuttal and the other reviews, I still think that the paper has significant contributions in SR, and will leave my original score unaltered. Well done!


Review 4

Summary and Contributions: This paper presents a novel super-resolution approach that exploits the cross-scale patch recurrence property of natural images. To exploit the patch recurrence property, the paper proposes two novel modules: the graph construction and graph aggregation modules. The graph construction module finds the k nearest neighbors in the feature space for each patch in the input image, while the graph aggregation module aggregates information from the k nearest patches. The two modules are adopted to an end-to-end network architecture called Internal Graph Convolutional Network to produce SR results. The paper also proposes adaptive patch normalization to resolve the discrepancy between nearest neighbors, e.g., their color and brightness, which is inspired by the adaptive instance normalization (AdaIN).

Strengths: While cross-scale patch recurrence has been utilized by traditional exemplar-based SR methods, this paper presents the first SR method that is based on deep learning. The proposed network architecture is reasonably designed and convincing. The evaluation and analysis are thorough and convincing.

Weaknesses: Unfortunately, the improvement is very marginal. In many cases, the proposed method achieves less than 0.1 dB higher PSNRs than previous methods, which makes the paper less interesting. While the evaluation includes a number of recent SR methods, it still does not entirely cover recent state-of-the-art methods such as the winning methods of the super-resolution challenges in AIM 2019 and NTIRE 2019 should be covered. Some details of the proposed network model are missing: the computation time and model size. All the evaluations are focused on non-blind cases, and it is not clear how the proposed method would work for images with unknown blur kernels.

Correctness: I think so.

Clarity: The paper is well-written and easy to understand. Typos & grammatical errors • 12: rely on ==> relying on ? • 62: vertexes ==> vertices • 112: the aggregated neighboring patches in the same scale of the query one which fails to incorporate the HR information and leads to a limited performance improvement for SISR. ==> a broken sentence... • 198: cross ==> across • 389: 2019 ==> In ICLR 2019

Relation to Prior Work: Yes

Reproducibility: Yes

Additional Feedback: ===== After rebuttal ==== After reading the rebuttal and the other reviews, I also raise my score to borderline accept. While I still have some doubt whether the performance improvement is significant enough, I think the proposed module is well-designed and interesting, and the performance gain on large-resolution images (Urban100, Manga109) is notable. What I still have a doubt about is that it is usually much easier to achieve performance improvement when the baseline is weak, and I suspect that the proposed method would not lead to much performance improvement with a better baseline model.

[Author Response · NeurIPS 2020]

We thank reviewers for the insightful comments. Overall, all reviewers noted the novelty and convincing results of
IGCN. Due to space limit, we only provide answers to main concerns. We shall fix minor issues and typos in the final
version. We will release our code once this work is accepted.

**R1: Comparison of same-scale aggregation and cross-scale aggregation.** Ta-
ble 2 in the submission shows that cross-scale aggregation (GraphAgg) performs
better than fully-connected same-scale aggregation (Non-local block). In Table 1,
we further report a baseline that finds and aggregates $k$ neighbors within the
same scale. Cross-scale aggregation still outperforms same-scale aggregation
by a considerable margin. We believe these results are adequate to show the

Table 1: Comparison of GraphAgg with same-scale and cross-scale aggregation on Urban100 ($\times 2$).

|  | Baseline (EDSR) | same-scale | cross-scale |
|---|---|---|---|
| PSNR | 32.93 | 33.01 | **33.23** |
| SSIM | 0.9351 | 0.9364 | **0.9383** |

effectiveness of our GraphAgg, i.e., aggregation across scales indeed obtain useful HR information. We will revise
the statement of "hardly improve" in L45-49 since same-scale aggregation also improves the baseline, despite being
marginal compared to cross-scale aggregation.

**R1: Difference from [34].** There are two main differences: 1) Different from
[34, 23, 28, 41] that exploit and aggregate recurrent patches within LR input
image, our method aggregates cross-scale internal HR cues and obtains an HR
feature $F_{L\uparrow s}$ directly by GraphAgg. Table 2 in the manuscript and Table 1
demonstrate the effectiveness of cross-scale aggregation. 2) We introduce AdaPN
that reduces the color discrepancy between query patch and $k$ neighbor patches,
keeping the high-frequency texture information unchanged. As shown in Table 6
in the manuscript and Figure 2 in the suppl., AdaPN allows more robust patch
aggregation and benefits the subsequent image restoration.

**R1: The relationship between performance gain and self-similarity level.** As
shown in Figure 1, our method performs better in regions with self-similarity,
especially in regions where texture patterns are extremely small. Besides, the
performance can also be well maintained to that of EDSR in regions with few
self-similar patches. More analysis will be provided in our final version.

Self-similarity Level     PSNR Gain

Figure 1: Examples to show the relationship between self-similarity level and PSNR gain (over EDSR). The brighter regions indicate larger values.

**R1+R4: Is IGCN dependent to the downsampling kernels? Does it work**
**for blind SR?** The patch matching for graph construction is performed in the
VGG feature domain, which is relatively robust for different degradation kernels.
Figure 2 shows an example of blind SR with an unknown blur kernel. IGCN
recovers sharper result than ZSSR and EDSR. Our result is better because IGCN
obtains and aggregates $k$ image-specific HR exemplars, which form helpful
internal complements when the blur kernel is unseen in the training dataset.

**R2: How about perceptual quality?** We compare our method with other SOTA
methods in terms of LPIPS, a perceptual quality metric for images, (AlexNet
version, Richard Zhang et al., CVPR'18). As shown in Table 2, IGCN achieves
the best LPIPS scores for all scale factors. Besides, the visual results provided in
the manuscript and suppl. also suggest the capability of IGCN in generating sharp and visually-pleasant images.

(a) LR      (b) ZSSR

(c) EDSR      (d) IGCN (ours)

Figure 2: Results of blind SR ($\times 4$).

**R2+R4: Running time.** We provide a runtime comparison in Table 5 in
the suppl. Benefit from the design of the searching window, IGCN runs
about two times faster than the SOTA method SAN.

**R3: What if similar patches are inexistent?** Glasner *et al.* in [9] (their
Figure 2) report that above 80% image patches exist 5 or more similar

Table 2: Comparisons on Urban100 in terms of LPIPS. (Lower scores indicate better.)

|  | RDN | RNAN | OISR | SAN | EDSR | IGCN |
|---|---|---|---|---|---|---|
| $\times 2$ | 0.0552 | 0.0579 | 0.0531 | 0.0541 | 0.0553 | **0.0520** |
| $\times 3$ | 0.1421 | 0.1440 | 0.1381 | 0.1392 | 0.1413 | **0.1375** |
| $\times 4$ | 0.2055 | 0.2037 | 0.2027 | 0.2031 | 0.2039 | **0.2006** |

patches across different scales. Even if there are discrepancies between found neighbors and query, AdaPN can align
neighbors to the query and reduce the low-frequency discrepancies. Moreover, ECN weights $k$ HR patches adaptively
for aggregation in accordance with difference between neighbor and query. ECN tends to output very small aggregation
weights for dissimilar neighbors. As such, errors caused by dissimilar neighbors are well suppressed in our network.

**R3: Do optimal values of $d$ and $k$ depend on the input resolution?** Due to the design of the searching window, the
input resolution will not affect the selection of optimal values. Regardless of input resolution, we search for $k$ neighbors
in a $d \times d$ window for aggregation. In addition, we select the optimal values of $d$ and $k$ on Urban100, which contains
images with different resolutions. Thus, the selected $d$ and $k$ work well for different resolutions.

**R4: Performance improvements are minor.** Our IGCN shows performance gain of 0.2~0.3dB over baseline EDSR
(which IGCN built upon) on large resolution benchmarks, i.e., Urban100 and Manga109. Although our performance
does not exceed the SOTA method by a large margin, we believe the PSNR gain over baseline and ablation results
(shown in Table 2 in the manuscript and Table 1) are adequate to show the effectiveness of our method. IGCN could
perform better accordingly if a better base model is employed.

**R4. Compare with winners of SR challenges AIM 2019 and NTIRE 2019.** The comparisons will be unfair. All
winners (i.e., ADCSR, IMDN, Efficient SR Network, ASSR) of AIM 2019 in different tracks adopt Flickr2K (2,650
images) as an additional training dataset to DIV2K (800 images). Differently, we follow the standard setting of main
conference papers, using only DIV2K for training. The two SR challenges (Real SR and video SR) in NTIRE 2019 are
different tasks with ours. Our comparisons already covered recent SOTA in CVPR'19 [5, 13, 21] and ICLR'19 [41].

[Meta-Review · NeurIPS 2020]

All four reviewers felt that the paper was above acceptance threshold. Therefore, it should be accepted.